# Imaging nanomagnetism and magnetic phase transitions in atomically thin CrSBr

Märta A. Tschudin [1,9], David A. Broadway [1,9] ✉, Patrick Siegwolf [1], Carolin Schrader[1], Evan J. Telford[2,3], Boris Gross[1], Jordan Cox [3], Adrien E. E. Dubois[1,4], Daniel G. Chica[3], Ricardo Rama-Eiroa [5,6], Elton J. G. Santos[5,6,7], Martino Poggio [1,8], Michael E. Ziebel [3], Cory R. Dean[2], Xavier Roy [3] & Patrick Maletinsky[1] ✉

Since their first observation in 2017, atomically thin van der Waals (vdW) magnets have attracted significant fundamental, and application-driven attention. However, their low ordering temperatures, $T_c$, sensitivity to atmospheric conditions and difficulties in preparing clean large-area samples still present major limitations to further progress, especially amongst van der Waals magnetic semiconductors. The remarkably stable, high-$T_c$ vdW magnet CrSBr has the potential to overcome these key shortcomings, but its nanoscale properties and rich magnetic phase diagram remain poorly understood. Here we use single spin magnetometry to quantitatively characterise saturation magnetization, magnetic anisotropy constants, and magnetic phase transitions in few-layer CrSBr by direct magnetic imaging. We show pristine magnetic phases, devoid of defects on micron length-scales, and demonstrate remarkable air-stability down the monolayer limit. We furthermore address the spin-flip transition in bilayer CrSBr by imaging the phase-coexistence of regions of antiferromagnetically (AFM) ordered and fully aligned spins. Our work will enable the engineering of exotic electronic and magnetic phases in CrSBr and the realization of novel nanomagnetic devices based on this highly promising vdW magnet.

Heterostructures based on 2D vdW materials have had a profound impact on our understanding and control of a vast range of electronic and optical phenomena in condensed matter physics[1–3]. However, explorations of 2D vdW magnets[4–8], which are typically highly fragile[9–11] and exhibit low ordering temperatures[4,5,12,13], are still in their infancy, despite remarkable progress in observing exotic forms of magnetism, including 2D-XY magnetism[14], orbital ferromagnetism[15], and Moiré magnetism[16]. The magnetically ordered vdW semiconductor CrSBr has

emerged as a highly attractive candidate to overcome these shortcomings[17,18]. Next to its in-plane, A-type AFM ordering (Fig. 1a), CrSBr exhibits unusual transport properties[19], an intriguing interplay between magnetic and optical properties[20], and a remarkable tunability of its magnetism by strain[21]. Importantly, and unlike other 2D magnetic vdW materials, CrSBr shows remarkable structural stability[22], magnetically orders at a relatively high (Néel) temperature $T_N \approx 132K$, with evidence for ferromagnetic (FM) intralayer interactions persisting

[1]Department of Physics, University of Basel, Basel, Switzerland. [2]Department of Physics, Columbia University, New York, NY, USA. [3]Department of Chemistry, Columbia University, New York, NY, USA. [4]QNAMI AG, Hofackerstrasse 40 B, Muttenz CH-4132, Switzerland. [5]Donostia International Physics Center (DIPC), 20018 Donostia-San Sebastián, Basque Country, Spain. [6]Institute for Condensed Matter Physics and Complex Systems, School of Physics and Astronomy, The University of Edinburgh, Edinburgh EH9 3FD, UK. [7]Higgs Centre for Theoretical Physics, The University of Edinburgh, Edinburgh EH9 3FD, UK. [8]Swiss Nanoscience Institute, University of Basel, Basel, Switzerland. [9]These authors contributed equally: Märta A. Tschudin, David A. Broadway. ✉e-mail: david.broadway@rmit.edu.au; patrick.maletinsky@unibas.ch

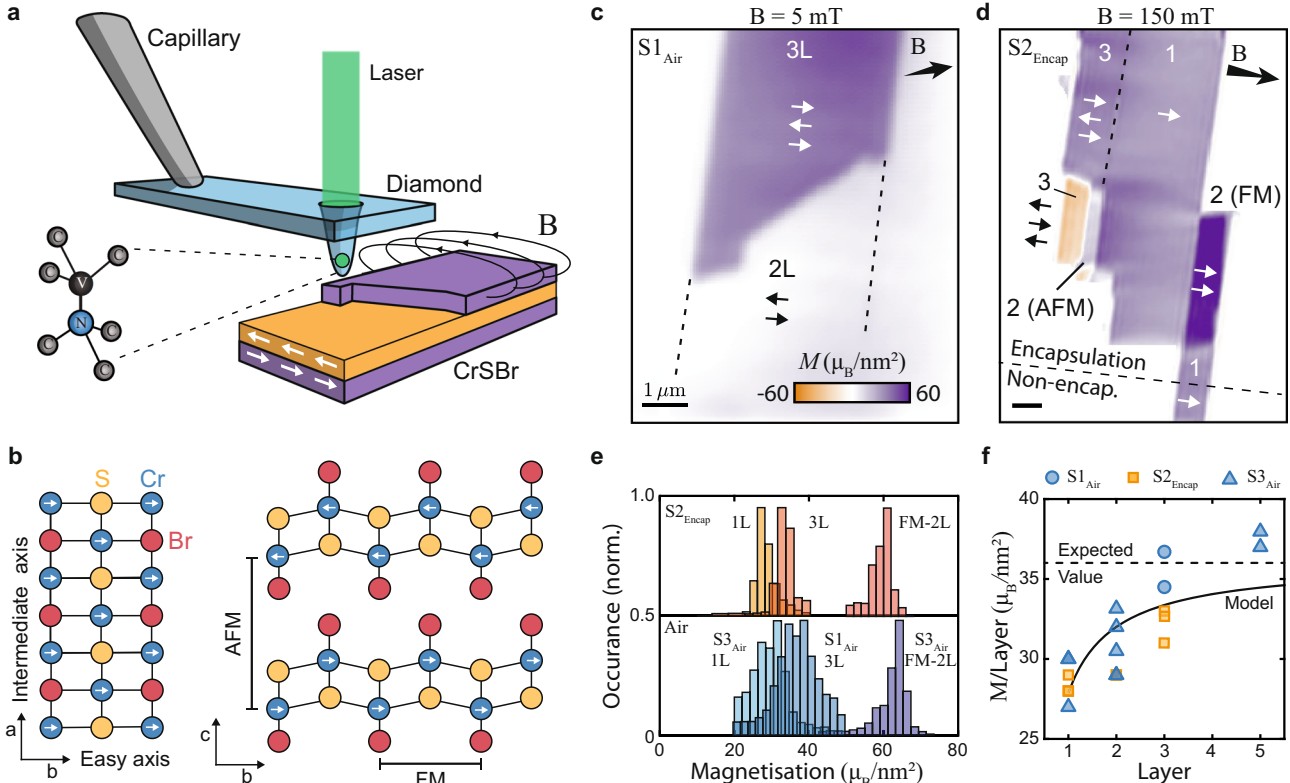

**Fig. 1 | Magnetic characterization of few-layer CrSBr. a** Illustration of the experiment involving an all-diamond tip containing a single-spin magnetometer that is scanned over the sample to spatially image magnetic stray fields. **b** Crystallographic structure of CrSBr, where the magnetic easy (intermediate) axis is aligned along the $b(a)$-axis of the crystal respectively. **c** Magnetization image of a non-encapsulated bilayer and trilayer flake (Sample 1, $S1_{Air}$), obtained in a low bias magnetic field $|B_{ext}|$ = 5mT. **d** Magnetization image of an encapsulated multi-layer flake (Sample 2, $S2_{Encap}$), obtained in a bias magnetic field $|B_{ext}|$ = 150mT strong

enough to induce FM ordering in some bilayer sections. **e** Exemplary magnetization histograms of encapsulated (top panel) and non-encapsulated (bottom panel) flakes of different thicknesses taken from additional datasets. **f** Extracted magnetization per CrSBr layer for encapsulated (orange) and non-encapsulated (blue) flakes as a function of the number of layers. The dashed line is the expected value (36 $\mu_B$/nm$^2$) from bulk measurements and the solid black line is a simplified model fit (see text).

to even higher temperatures[17,22,23]. Yet, while recent studies addressed both thick[24] and thin[25] CrSBr flakes on the microscale, little is thus far known about the nanoscale properties of thin CrSBr and how its various magnetic phases develop and transition in few-layer thin samples, which have so far been addressed only by non-quantitative[23,26] and invasive[27] imaging methods.

Here, we employ single-spin scanning magnetometry[28] using an individual nitrogen-vacancy (NV) center in diamond[29] – a nanoscale imaging technique that is non-invasive and sensitive enough to image magnetism in vdW monolayers[12]—to explore magnetic order in CrSBr in a quantitative way. The NV spin is situated at the tip of an atomic force microscope and scanned across the CrSBr flakes (Fig. 1a) to quantitatively image magnetic stray fields through Zeeman shifts of the spin's energy levels. The NV thereby enables a quantitative[30] determination of $B_{NV}$—the projection of the total magnetic field onto the NV axis—which in turn allows us to determine CrSBr's sample magnetization and magnetic anisotropy energy, down to the monolayer limit. Furthermore, building on the high stability and spatial resolution of our approach, we provide direct visualizations of key magnetic phase transitions in mono- and bilayers of CrSBr.

## Results

We start by investigating few-layer CrSBr samples at $T \approx 4$K and at low magnetic fields ($B \approx 5$ mT), where the CrSBr layers exhibit AFM interlayer alignment[17], and where a nonzero net sample magnetization can thus only be expected from odd numbers of layers. Our samples were fabricated by mechanically exfoliating bulk CrSBr crystals onto a Si/

SiO$_2$ substrate, where a subset of the resulting flakes were encapsulated in hexagonal boron nitride (hBN) and the rest remained bare, non-encapsulated flakes. CrSBr typically exfoliates into large, near-rectangular flakes, where the long (short) edges of the rectangle correspond to the $a$ ($b$) crystallographic axes (Fig. 1b). Axis $b$ corresponds to the magnetic easy-axis, where magnetization alignment along $a$ is suppressed by an energy penalty that has not been determined for few-layer samples thus far.

We perform scanning NV magnetometry across our CrSBr flakes to obtain magnetic images, from which we reconstruct the underlying magnetization structure using a neural network approach[31]. Figure 1c, d shows representative data on flakes containing mono-, bi-, and tri-layer sections of CrSBr. Throughout, these data reveal largely uniform magnetizations, which are well-aligned with the magnetic easy axis $b$.

We determine the average magnetization of CrSBr flakes through histograms of the reconstructed magnetization maps (Fig. 1e). Applying this approach to a series of CrSBr flakes with increasing layer numbers, we observe an initial increase in the magnetization per layer, until reaching the expected saturation magnetization at $N \gtrsim 5$ layers to the expected bulk value of 36 $\mu_B$/nm$^2$ (Fig. 1f), where $\mu_B$ is the Bohr magneton, additional data supplementary information (SI) Fig. 14. To validate our findings, we independently determined CrSBr sample magnetizations by performing analytic fits to the measured magnetic field emerging from flake edges, which agrees with our initial approach (see methods sections for details). The observed increase of magnetization with the number of layers suggests that interlayer exchange coupling[32] plays a relevant role in stabilizing intralayer FM ordering.

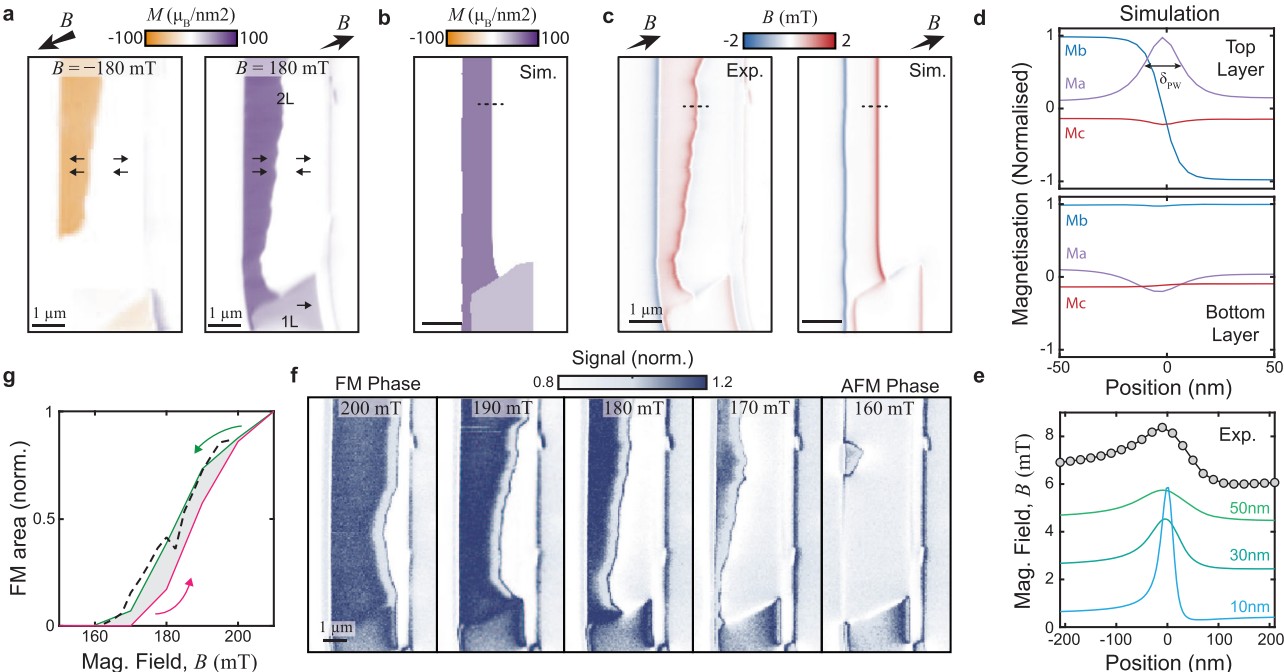

**Fig. 2 | Imaging the spin–flip transition in bilayer CrSBr. a** Image of coexisting regions of AFM and fully magnetized phases in a CrSBr bilayer obtained at $|B_{ext}| = \pm 180$mT after demagnetizing the sample from higher fields, with a magnetic field axis oriented along $(\theta, \phi) = (53°, 16°)$ (see text). The images were obtained by magnetization reconstruction from raw $|B_{NV}|$ data. **b** Micromagnetic simulation of the phase wall (PW) under the same conditions as the right panel in (**a**). **c** Experimental magnetic field image from the measurement in (**a**) and simulated magnetic field from (**b**). **d** Cross sections of the magnetization vector across the PW

(dashed line in **b**) for both the top and bottom layers. **e** Comparison of the experimental measured magnetic field line cut across the PW (dashed lines in **c**) and the simulation with various NV-sample standoff distances, offset for clarity. **f** Qualitative NV magnetometry images of the movement of the PW when demagnetizing the sample from the fully magnetized to the AFM state. **g** Magnetic hysteresis of the bilayer sample, calculated as a ratio of the total area in the fully magnetized state versus the AFM state. The dashed line corresponds to data from an additional measurement run.

Indeed, the scaling of layer magnetization, $M_l$, with $N$ fits well to an empirical model (black line in Fig. 1f) that assumes a magnetization reduction, $\epsilon$, for the outermost layer of Cr atoms, but constant magnetization, $M_0$, for the remaining Cr layers $M_l = M_0(N - 2\epsilon)/N$. The model yields a bulk magnetization $M_0 = 36.1(1)\mu_B/$nm², and a reduction factor of 11(4)% for the outermost layers. We note that our findings are reproduced and near-identical whether we investigate encapsulated or non-encapsulated flakes (Fig. 1f). This attests to the remarkable air-stability of magnetism in CrSBr, which we observe down to monolayers that had been left exposed to ambient conditions for days. While we have not conducted an in-depth analysis of monolayer CrSBr's resilience to air, this already indicates that it has a remarkable improvement in ambient stability when compared to similar semiconductor magnets e.g. chromium and vanadium trihalides.

For magnetic fields of a few 100 mT applied along the $b$-axis, CrSBr undergoes a metamagnetic transition from an AFM state, with low magnetization, to a state of strong magnetization. Given the strong $b$-axis anisotropy of CrSBr, this metamagnetic transition was identified as a spin–flip transition, wherein the high-field phase, spins in adjacent layers all align in a ferromagnetic configuration[21,22]. Past observations of non-symmetric hysteresis curves around the spin–flip transition[33], suggest the possibility that the two phases of zero and nearly saturated magnetization can coexist during the transition. However, whether such a mixed phase indeed exists and how the spin–flip transitions occur on the microscopic level in CrSBr remains unknown.

To address the physics of this spin–flip transition and phase-coexistence, we initialize a CrSBr bilayer into the FM configuration by applying a magnetic field $B_{ext} = 230$ mT along the NV axis, such that the in-plane projection of $B_{ext}$ is approximately aligned with the sample's $b$-axis. The field's polar and azimuthal angles amount to $(\theta, \phi) = (53°, 16°)$, where $\theta = 0°$ corresponds to the sample normal and

$\phi = 0°$ to the horizontal axis in all images. We subsequently decrease $B_{ext}$ in 10mT steps and perform magnetic imaging to identify the spin–flip field. We observe that at $B_{ext} \approx 200$ mT, an AFM-ordered region develops in the flake. Figure 2a shows a full magnetization map obtained at $B_{ext} \approx 180$mT that evidences the coexistence of phases with zero and nearly saturated magnetization during the spin–flip transition. This observation is reproducible and analogously occurs at inverted magnetic fields. The boundary between the two phases that we term a "phase wall" (PW) is stable over several days, which is consistent with the spin–flip transition exhibiting a large imaginary AC magnetic susceptibility that implies irreversible domain wall movement (see SI Section 3). The weak, nonzero stray field that is observed on the right side of the flake (Fig. 2a) is attributed to either a small degree of spin canting towards the edge of the flake, or a small stripe of an odd-layer CrSBr adjacent to the bilayer.

To provide a deeper understanding of the mechanism behind the formation of the PW, we performed micromagnetic simulations, taking into account the known magnetic properties of CrSBr[34–36] (details in SI Section 4). For a homogeneous CrSBr bilayer, we are unable to stabilize a PW in our simulations, i.e. the flake immediately switches from AFM to a strongly magnetized state (or vice versa) within one field step at 160 mT (195 mT). We conjecture that inhomogeneities in the sample contribute to the stabilization of the PW. We model these by imposing that the interlayer exchange coupling strength $A_{ex,z}$ exhibits a linear variation along the $b$-axis that amounts to 24% across the flake width of 3.3$\mu$m. While other inhomogeneities or pinning centers could also explain the stability of the PW, a variation of $A_{ex,z}$ appears plausible, as it can be induced by strain gradients[21] that commonly occur in van der Waals structures[37]. Using literature values[21], we estimate that the variation in $A_{ex,z}$ we impose can be induced by a realistic 0.4% strain variation across the CrSBr flake. Figure 2b, c shows the resulting

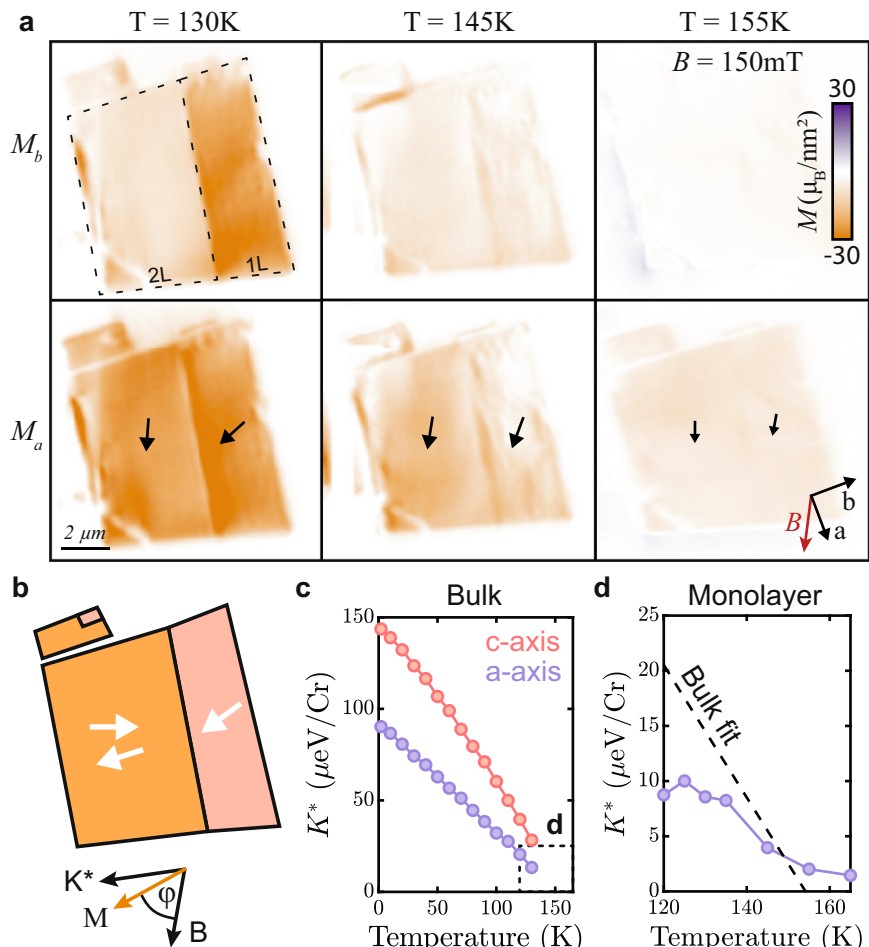

**Fig. 3 | Determination of magnetic anisotropy in a CrSBr monolayer.**
**a** Reconstruction[31] of magnetization components along the CrSBr $b$-axis ($M_b$, top panels) and $a$-axis ($M_a$, bottom panels) for different temperatures, under a magnetic bias field applied roughly along the $a$-axis (($\theta$, $\phi$) = (65°, 265°)). **b** Illustration of the two competing interactions: effective anisotropy ($K'$) and Zeeman energy ($B$), and their effect on the magnetization direction of the material. **c** Effective in-plane and out-of-plane anisotropy of bulk CrSBr measured with vibrating sample magnetometry. **d** Extracted effective in-plane anisotropy ($K'$) from the measurements in (**a**) and the extrapolated anisotropy from the bulk measurements in (**c**) assuming a linear trend. The dashed box in (**c**) indicates the range of (**d**).

simulated sample magnetization and stray field pattern in the presence of the PW, where the latter shows good agreement with our experimental data.

Our simulations further show that the magnetization rotation across the PW is largely confined to the $a$-$b$ plane, with a sense of rotation that is set by the nonzero projection of $B_{ext}$ onto the $a$-axis. The simulated PW extends over a (Bloch) width of $\delta_{PW}$ = 18 nm (Fig. 2d), which is below the spatial resolution of our method. Yet, the experimental data show good agreement with the model and their comparison allows us to estimate the NV-sample distance (that sets the spatial resolution) to ~50 nm (Fig. 2e).

To further investigate the spin–flip transition in bilayer CrSBr, we follow the movement of the PW through the flake as the spin–flip transition occurs. For this, we use a qualitative "dual-iso-B" imaging modality (details in methods section) that allows for faster imaging than the fully quantitative method employed so far. Such "dual-iso-B" imaging yields a magnetic field-dependent signal and with it, an efficient method to determine changes in the PW position. We thereby obtain the series of images (3 hours per image) shown in Fig. 2f that evidences the stability and incremental motion of the PW through the sample as $B_{ext}$ is decreased (see SI Section 1 for additional data and information). This smoothness is further evidence of only weak magnetic pinning in CrSBr. From these data, we extract hysteresis curves (Fig. 2g) by determining the relative bilayer areas in the AFM and

strongly magnetized state, as a function of $B_{ext}$ (see SI Section 1c). The resulting curve is consistent with previous, macroscopic measurements[33], which supports the notion that past hysteresis measurements in the CrSBr spin–flip transition indeed are explained by PW movement throughout the sample.

We now examine the properties of few-layer CrSBr above and below its magnetic ordering temperature and begin by exploring the material's magnetic anisotropy as a function of temperature. For this, we study adjacent mono- and bilayer flakes, where we apply a bias magnetic field $B_{ext}$ = 150 mT along the intermediate $a$-axis, to tilt the magnetization away from its easy axis, $b$. Using our reconstruction method[31] we determine the magnetization both along the $a$-axis ($M_a$) and $b$-axis ($M_b$) simultaneously (Fig. 3a). For the bilayer, we observe that the magnetization is aligned with the applied field for all temperatures and steadily decreases with $T$. This observation is readily explained by spin-canting out of the AFM phase and a decrease of the magnetization as $T$ approaches $T_N$. In contrast, the monolayer magnetization for our initial temperature $T$ = 130 K is initially nearly aligned with the easy axis (Fig. 3b), indicating that the effective magnetic anisotropy energy, $K'$, exceeds the Zeeman energy. As we increase $T$, the monolayer magnetization not only decreases in magnitude but also reorients towards the $a$-axis (see methods section and SI Figs. 15, 17). This reorientation results from a reduction of $K'$ with increasing temperature due to thermal-magnon-induced softening of the

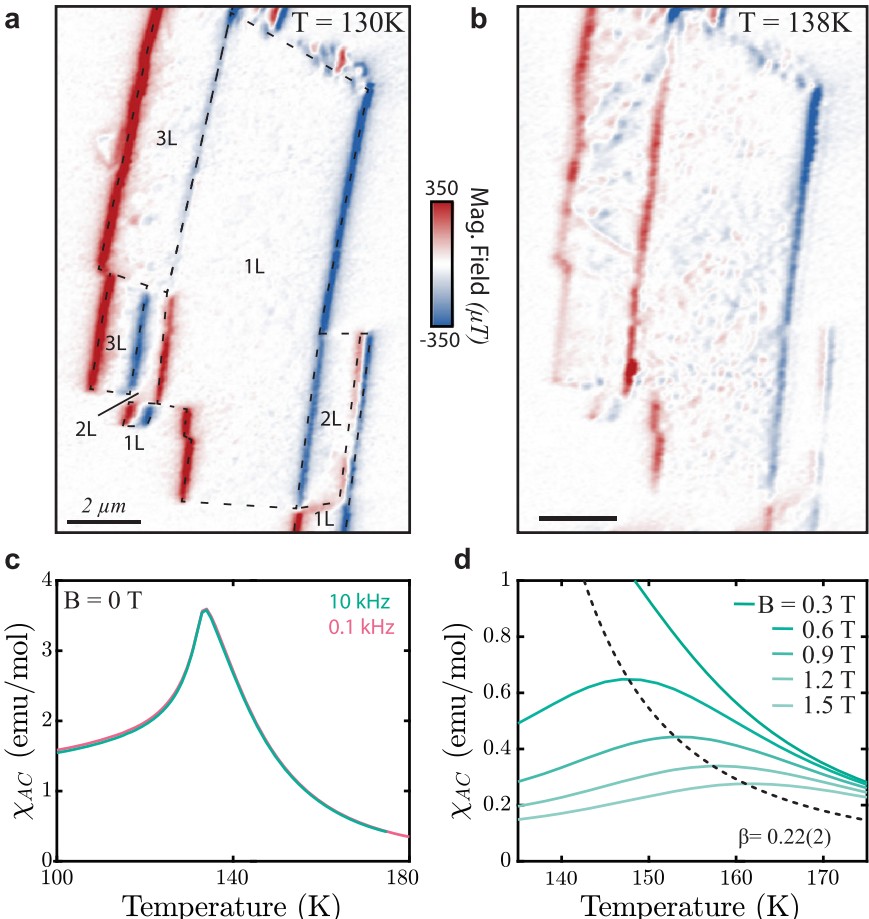

**Fig. 4 | Properties of CrSBr monolayer near $T = T_N$. a** Magnetic field image of Sample $S2_{Encap}$ at $T = 130K$ with a bias magnetic field ($B_{ext} = 5mT$ applied along the b-axis). **b** Same as (**a**), taken at $T = 138K$. The onset of short-range magnetic inhomogenities is apparent in regions with odd numbers of layers. **c** Real part of the zero-field, AC magnetic susceptibility $\chi_{AC}$ of bulk CrSBr. The data indicate a lack of additional magnetic phase transitions for $T > T_N = 132K$. **d** Same as (**c**) obtained in nonzero bias magnetic fields $B_{ext}$. From these data, we extract critical parameters $\beta = 0.22(2)$ and $\gamma = 2.28(2)$, both approaching the expected values for an ideal 2D-XY spin model ($\beta = 0.231$ and $\gamma = 2.4$[14]). AC susceptibility data were collected with $B$ parallel to the $a$-axis with an oscillating field of 1mT.

ferromagnet[38] (Fig. 3c). Using the experimentally determined, temperature-dependent magnetization and canting angle of the monolayer flake, we directly and quantitatively extract $K^*$ for monolayer CrSBr (Fig. 3d, see methods section). Extrapolating the bulk anisotropy measurement shown in Fig. 3c to higher temperatures (Fig. 3d, dashed line) yields qualitative agreement with our results and indicates that the primary source of magnetic anisotropy for the monolayer is magnetocrystalline anisotropy. While our measurement of the monolayer anistropy is consistent with the bulk measurements in both magnitude and critical behavior, it deviates from the scaling as a function of temperature. While we do observe a reduction in magnetization in the few-layer limit, this is not enough to explain this potential difference. As such, this may indicate that the proportionality of anisotropy to magnetization in the thin limit is fundamentally different from bulk. While we do not address this possibility here, we are hopeful that future research will explore this.

Our data in Fig. 3 indicate that FM order in the monolayer persists well beyond the nominal bulk Néel temperature $T_N \approx 132K$ of CrSBr, consistent with earlier observation of intralayer correlations in CrSBr for $T > T_N$[22,27]. To further elucidate the still unclear nature of this intermediate FM regime[23], we investigate a monolayer CrSBr flake (Fig. 4a) in a temperature range slightly above $T_N$ and at low magnetic fields. We note that due to microwave heating the temperature measured constitutes a lower bound for the sample temperature, which is likely a few Kelvin higher than the indicated temperature values – see

SI for more details. We observe that once $T \gtrsim T_N$, the monolayer develops sizable spatial variations of magnetization (Fig. 4b) that are not observed once $T \lesssim T_N$. The spatial gradient in the magnetization variation that is visible in Fig. 4b is tentatively assigned to a gradient in temperature resulting from heating due to the microwave control line, used to drive the NV, which was located on-chip and near the left edge of the image. We also observed such inhomogeneities on trilayer flakes, whereas even-numbered layers do not exhibit any noticeable inhomogeneity (see bilayer sections in Fig. 4a, b and additional data in SI Section 2). The latter suggests that weak remaining AFM interlayer coupling above $T_N$ can still compensate for the observed inhomogeneities.

Further experimental observations help us identify the most likely origin of the inhomogeneities in the intermediate FM phase. First, we note that the inhomogeneities appear completely static over the timescale of our experiments of several days. This excludes critical spin fluctuations near $T_c$, or the appearance of unbound meron/anti-meron pairs[26] as their origin. Second, a near-homogenous monolayer magnetization can be restored by increasing $B_{ext}$ to around 200mT (see SI Fig. 7), indicating that the saturation magnetization is nearly constant across the sample. Given that the magnetic anisotropy $K^*$ is near-vanishing for $T \gtrsim T_N$ variations in $K^*$ appear unlikely as the origin for the observed inhomogeneities. This would leave local variations of intralayer exchange couplings as the most likely explanation for our observation. These local variations could originate from atomistic

defects, strain variation, or other local disorders that could disrupt the regular crystallographic structure.

To further elucidate the nature of magnetic ordering in the intermediate magnetic phase of CrSBr, we turn to measurements on bulk crystals. In Fig. 4c, we show the real part of the zero-field AC susceptibility $\chi_{AC}$ of a bulk CrSBr crystal, the imaginary part of $\chi_{AC}$ remains ≈ 0 over the entire temperature range studied here. Within the probed parameter range, $\chi_{AC}$ is frequency independent and displays a single cusp at $T_N = 132$K. The absence of an additional higher temperature feature in $\chi_{AC}$ indicates that, despite the presence of FM correlations persisting to temperatures $T > T_N$, no short- or long-range intralayer order emerges in CrSBr above $T_N$ at zero-field. However, in the presence of applied bias magnetic fields, a field-dependent maximum in $\chi_{AC}(T)$ emerges (Fig. 4d). The magnetic field dependence of this high-temperature feature can be used to extract critical exponents for magnetic ordering in CrSBr[39,40]. Intriguingly, the extracted values of $\beta = 0.22(2)$ and $\gamma = 2.28(2)$ are close to the values expected for the 2D-XY model ($\beta = 0.231$ and $\gamma = 2.4$[14,41]). The value of $\beta$ we determined aligns with values previously determined by a range of experimental techniques[42,43] and earlier conclusions that CrSBr follows 2D-XY-like behavior above $T_N$[26].

Our combined findings indicate that easy-plane behavior of decoupled FM layers (e.g. 1L, 3L, ..., etc) could exist in a narrow temperature range above $T_N$ and below $T_C$, in which 2D-XY-like physics could thereby be observed in CrSBr. However, in our present samples, this behavior is likely masked by the inhomogeneities in intralayer exchange couplings, as discussed earlier. Future advances in material purification[44] and isolation from the substrate could therefore offer exciting perspectives for observing 2D-XY spin physics[45] with its accompanying topological spin textures[26] in monolayers of the CrSBr family.

In this work, we provided a quantitative, nanoscale study of magnetization strength and anisotropy in a few layer CrSBr samples and addressed two key magnetic phase transitions by direct magnetic imaging using single-spin magnetometry. We observed that the magnetization per CrSBr layer decreases monotonically with layer number, but remains nonzero even for monolayers, which we find magnetically stable even in the absence of encapsulation. We further investigated the AFM–FM spin–flip transition in bilayers and found them to be driven by the nucleation and subsequent propagation of a phase wall, rather than a coherent rotation of layer magnetization. Finally, we addressed the evolution of magnetization near CrSBr's critical temperature and directly evidenced the reduction of anisotropy when approaching $T_N$. Near $T_N$, we observed the onset of magnetic inhomogeneities in odd-layer flakes, which we attributed to local disorder in intralayer exchange couplings that currently mask a native 2D-XY behavior of CrSBr monolayers.

Our results underline CrSBr's significant potential for the development of novel technologies based on 2D magnets. In particular, the air stability, and large-range uniformity of magnetization across tens of microns evidence robustness and scalability of this material, while the highly stable magnetic PWs we discovered suggest potential interesting functionalities in the context of spintronics and racetrack memory devices[46].

## Methods
### Sample fabrication
**Synthesis of CrSBr bulk crystals.** Large single crystals of CrSBr were grown using a chemical vapor transport reaction described in Scheie et al.[42].

**Fabrication of non-encapsulated CrSBr samples.** CrSBr flakes were exfoliated onto 90 nm $SiO_2/Si^+$ substrates (NOVA HS39626-OX9) using mechanical exfoliation with Scotch® Magic™ tape[47,48]. Before exfoliation, the substrates were cleaned with a gentle oxygen plasma to

remove adsorbates from the surface and increase flake adhesion[49]. The exfoliation was done under ambient conditions. Flake thickness was identified using optical contrast and then confirmed with atomic force microscopy[22,23].

**Fabrication of encapsulated CrSBr samples.** CrSBr flakes were exfoliated onto 285 nm $SiO_2/Si^+$ substrates (NOVA HS39626-OX) using mechanical exfoliation with Scotch® Magic™ tape[47,48]. Before exfoliation, the substrates were cleaned with a gentle oxygen plasma to remove adsorbates from the surface and increase flake adhesion[49]. The exfoliation was done under inert conditions in an $N_2$ glovebox with <1 ppm $O_2$ and <1 ppm $H_2O$ content. Thin flakes (<10 nm thick) of hexagonal boron nitride (hBN) were then placed on top of the desired CrSBr flakes to encapsulate them using the dry-polymer transfer technique[50]. CrSBr flake thickness was identified using optical contrast and then confirmed with atomic force microscopy after encapsulation with hBN[22,23].

**Atomic force microscopy for sample screening.** Atomic force microscopy was performed in a Bruker Dimension Icon® using OTESPA-R3 tips in tapping mode. Flake thicknesses were extracted using Gwyddion to measure histograms of the height difference between the substrate and the desired CrSBr flake.

### NV experimental apparatus
The magnetic images were taken using two different Attocube cryostats, an attoLIQUID 1000 for measurements around $T = 4$K and an attoDRY 2200 for variable temperatures $T = 2-300$K measurements (see SI Fig. 10). The NV electronic spin is excited using a 532 nm laser (attoLiquid–Laser Quantum 532 Gem, and attoDry–Laser Quantum 532 Torus) and then the photoluminescence (PL) of the NV spin is separated using a dichoric mirror which is detected with an avalanche photodiode (Excelitas SPCM-AQRH-33). The microwave control is performed using a signal generator (attoLIQUID–SRS SG384, attoDry–Rhode Schwartz SMBV100B) which is applied to the NV using a wire that is bonded across the sample.

### Bulk measurements
All bulk magnetic measurements were performed on a Quantum Design Dynacool Physical Property Measurement System (PPMS). For oriented magnetic measurements, two single crystals of CrSBr were mounted on a quartz paddle using GE varnish with the a-, b-, or c-axis aligned with the direction of the instrument's magnetic field. For subsequent orientations, the varnish was removed using a 1:1 mixture of toluene and ethanol, and the same crystals were re-mounted in a different orientation. DC and AC magnetic measurements were performed with the PPMS vibrating sample magnetometry and AC magnetometry modules, respectively.

### NV measurement techniques
Various NV magnetometry techniques are used in this work. The primary is a pulsed or continuous wave optically detected magnetic resonance (ODMR)[51]. To extract the central NV frequency when using this method, each pixel is fitted with a Lorentzian function.

An alternative measurement technique involves frequency modulation of the MW to tune the applied frequency rapidly. This involves modulating the applied frequency using an IQ modulator where the frequency is shifted to a lower frequency using the functions:

$$I = \cos(t\Delta f/2) \tag{1}$$

$$Q = \sin(t\Delta f/2) \tag{2}$$

**Table 1 | Description of samples used in this study**

| Sample # | Label | Layers | Encapsulated |
|---|---|---|---|
| 1 | $S1_{Air}$ | 2, 3, 5 | No |
| 2 | $S2_{Encap}$ | 1, 2, 3 | Yes |
| 3 | $S3_{Air}$ | 1, 2, 4, 5 | No |
| 4 | $S4_{Air}$ | 1, 2 | No |

and to higher frequency

$$I = \sin(t\Delta f/2) \qquad (3)$$

$$Q = \cos(t\Delta f/2) \qquad (4)$$

where $t$ is time, and $\Delta f$ is the desired splitting of the two peaks which is typically set to the width of the ODMR feature. By normalizing these two signals such that

$$S = \frac{ODMR_1}{ODMR_2} \qquad (5)$$

we obtain a single spectrum that can be locked onto using a PID, see SI Fig. 11 for more details.

In this work, we use this signal in two different ways. The first is to perform a single frequency measurement, referred to as Iso-B[30], which qualitatively (quantitative in the case of small field changes and consistent standoff) images the magnetic field at the cost of dynamic range (Used in Fig. 2f). The other technique is to lock onto the zero signal point using a PID loop, which is slower to allow the PID to track the frequency but is quantitative, which is used throughout the manuscript.

### Sample characterization

In this work, we studied various different CrSBr samples, as summarized in Table 1. The samples contain different layer thicknesses of CrSBr ranging from monolayer to five layers. To assess the layer composition of the CrSBr flakes studied in this work we optically characterized the individual samples as shown in SI Fig. 12. The layer thickness of the CrSBr samples was determined by the relative optical contrast between the CrSBr flakes and a SiO2 layer. A single layer of this material has an absorption of approximately 15% in our optical microscope. The absorption scales linearly for thin layers (<3) and is less accurate for thicker layers (>5). For the encapsulated flakes of sample $S2_{Encap}$ additional AFM measurements were taken of the flake after encapsulation to independently verify the layer thicknesses.

### Layer-dependent magnetization estimation

To determine the magnetization of the CrSBr flakes we use two separate methods: The first method is fitting the magnetic stray field of a 1D line cut across the edge of a flake and the second method is taking a histogram of the reconstructed magnetization image of a CrSBr flake.

The first method we used to extract magnetization from our measurements is by fitting the magnetic field of a 1D line across the edge of the material. The magnetic field from a stray edge with an arbitrary magnetization direction is given by:

$$B_x = -\frac{\mu_0 M}{2\pi} \frac{z_{NV} \cos(\theta_M) + (x-x_0)\sin(\theta_M)\cos(\phi_M)}{z_{NV}^2 + (x-x_0)^2}, \qquad (6)$$

$$B_z = \frac{\mu_0 M}{2\pi} \frac{-z_{NV}\sin(\theta_M) + (x-x_0)\cos(\theta_M)\cos(\phi_M)}{z_{NV}^2 + (x-x_0)^2}, \qquad (7)$$

where $\mu_0$ is the vacuum permeability, $M$ is the magnetization magnitude of the material, $z_{NV}$ is the stand-off distance from the NV to the sample, $x$ is the position vector and $x_0$ is the position of the sample edge, $\theta_M$ is the angle of the magnetization from the z-axis, and $\phi_M$ is the azimuthal angle of the magnetization. The values for the NV angles $\theta_M$ and $\phi_M$ were independently determined in our experiments and are kept fixed during the fitting routine. Examples of these linecuts and their respective fits are shown in SI Fig. 13. The extracted magnetization and NV stand off distance are summarized in SI Table 1.

The second method of determining the magnetization is to use a neural network to reconstruct a magnetization image and then take a histogram of the image to determine the average magnetization of a given region. Examples of these histograms are shown in SI Fig. 14. Both methods for determining the magnetization give a similar values of magnetization (SI Table 1).

### Magnetization reconstruction

The reconstruction of magnetization from a magnetic field can be performed using a Fourier space transformation where the transformation from $\mathcal{B}$ to $\mathcal{M}$ is given by a transformation matrix $\mathcal{A}$ such that,

$$\mathcal{B} = \mathcal{A}\mathcal{M} \qquad (8)$$

where $A$ is non-invertible but is often approximated to retrieve a reliable reconstruction[52]. In the case of in-plane magnetization, such as in CrSBr, this reconstruction is difficult to perform as noise in the data often results in artifacts that are difficult to remove and an inconsistent estimation of magnetization strength. Likewise, in traditional reconstruction the magnetization direction is set in the reconstruction process, meaning that the magnetization can only vary along one direction. This has two problems, first, the magnetization direction is often not known and thus approximations of the direction can lead to further inconsistencies, and second, the magnetization direction may vary across the image.

To overcome these constraints we recently developed a machine learning approach that estimates the inversion transformation matrix ($\mathcal{A}^{-1}$) to produce a magnetization image[31]. This image is then transformed back into a magnetic field using the well-defined matrix $\mathcal{A}$ which is then compared with the measured magnetic field as an error function. Through this process, the magnetic image is fitted using the machine learning neural network, where the end result must be a valid solution to the measured magnetic field. That is, the error is minimized between the measured magnetic field and the projected magnetic field from the reconstructed magnetization. Additionally, it allows for a reliable estimation of the uniform magnetization direction of the material for arbitrary direction.

This previous work was still limited by the requirement that the magnetization direction needed to be uniform across the image. In this work, we were able to initialize the material such that the bi-layer, which had a low anisotropy, was orientated along the magnetic field direction, while the monolayer, which had a high anisotropy, was pointed along its preferred direction. In order to reconstruct this non-uniform magnetization direction we modified the neural network from previous work to have two channels. That is, rather than generating a magnetization image with a given magnetization direction $\theta_M$, it produces two images one for $M_x$ and one for $M_y$. These images can have their respective magnetic fields calculated in the same manner as previously and then the magnetic fields are combined to compare with the measured magnetic image. Furthermore, this approach allows for the addition of a mask that prohibits the attribution of magnetization to regions that are known to not host any magnetic material. A Schematic of the neural network is shown in SI Fig. 16 and the full reconstruction of all the temperature data is shown in SI Fig. 17.

## Estimation of monolayer anisotropy

In order to make an estimate of the in-plane anisotropy using the temperature series we use the Stoner-Wohlfarth model[53], as outlined in the following. We model the energy of a magnetic system under the influence of a magnetic field using the expression

$$E = KV\sin^2(\phi - \theta) - \mu_0 M_s V B_{ext}\cos\phi, \qquad (9)$$

where K is the magnetocrystalline anisotropy, V is the magnet volume, $M_s$ is the saturation magnetization, $B_{ext}$ is the external magnetic field, $\phi$ is the angle between the magnetization direction and the external field, and $\theta$ is the angle between the applied field and the easy axis of the material. See SI Fig. 15 for more details.

To analyze our data on monolayer CrSBr, we extract the anisotropy from the angle of the magnetic field and the measured magnetization direction, using the derivative of Eq. (9) with respect to the angle $\phi$,

$$\frac{\partial E}{\partial \phi} = 2KV\sin(\phi - \theta)\cos(\phi - \theta) + M_s V B_{ext}\sin\phi. \qquad (10)$$

Then we evaluate when the derivative is zero and rearrange to get,

$$K = \frac{M B_{ext}\sin\phi}{2\sin(\phi - \theta)\cos(\phi - \theta)}. \qquad (11)$$

Using this equation we determine the anisotropy of the monolayer using the measured magnetization M and direction $\phi$.

## Bulk measurements of anisotropy

Field-dependent magnetization measurements were collected for CrSBr along the a, b, and c axes at fixed temperatures between 2 to 130 K and in the magnetic field range 0 to 9 T. For a- (intermediate) and c- (hard) axis measurements, linear fits were applied to the low-field and high-field magnetization data. The intercept of these two linear fits was taken as the saturation field, and the y-intercept of the high-field linear fit was taken as the saturation magnetization ($M_{sat}$). For b-axis measurements, the low-field fit was replaced by a fit to the linear region at the metamagnetic transition field, and the saturation field and magnetization were determined in the same way as for the a and c axes. The temperature-dependent anisotropy field ($H_{ani}$) for the a (c) axis was then determined as the difference between the a- (c-) axis saturation field and the b-axis saturation field. The effective magnetic anisotropy energy ($K^*$) for the a and c axes was then calculated at each temperature using the Stoner-Wohlfarth model[53,54]:

$$K^* = \frac{\mu_0 H_{ani} M_{sat}}{2} \qquad (12)$$

## Data availability

The data generated in this study have been deposited in the Zenodo database under accession code https://doi.org/10.5281/zenodo.11375831.

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

## Acknowledgements

The authors acknowledge financial support from the ERC consolidator grant project QS2DM, by SNF project No. 188521, and from the National Center of Competence in Research (NCCR) Quantum Science and Technology (QSIT), a competence center funded by the Swiss National Science Foundation (SNF). Synthesis of the CrSBr crystals was funded by the Columbia MRSEC on Precision-Assembled Quantum Materials (PAQM) under award number DMR-2011738 and the Air Force Office of Scientific Research under grant FA9550-22-1-0389. Bulk magnetic measurements were supported under Energy Frontier Research Center on Programmable Quantum Materials funded by the US Department of Energy (DOE), Office of Science, Basic Energy Sciences (BES), under award DE-SC0019443. The instrument used to perform these magnetic measurements was purchased with financial support from the National Science Foundation through a supplement to award DMR-1751949. EJGS acknowledges computational resources through CIRRUS Tier-2 HPC Service (ec131 Cirrus Project) at EPCC funded by the University of Edinburgh and EPSRC (EP/P020267/1); ARCHER UK National Super-computing Service (http://www.archer.ac.uk) *via* Project d429. E.J.G.S. acknowledges the EPSRC Open Fellowship (EP/T021578/1), and the Edinburgh-Rice Strategic Collaboration Awards for funding support. Micromagnetic calculations were performed at sciCORE (http://scicore.unibas.ch/) scientific computing center at University of Basel. B.G. acknowledges the support of the Canton Aargau.

## Author contributions

The NV measurements were performed by M.A.T. and D.A.B. together with C.S. and P.R., under the supervision of P.M. D.A.B., M.A.T., and C.S. analyzed the Data. The Samples were prepared by E.J.T. and J.C., and the bulk crystal was synthesized by DGC, all under the supervision of X.R. and C.R.D. M.E.Z. performed bulk AC magnetic measurements and data analysis. B.G. and M.P. provided micromagnetic simulation of the material while RRE and EJGS provide theoretical description of the formation of the phase boundary. D.A.B., M.A.T., and P.M. wrote the manuscript. All authors discussed the data and commented on the manuscript.

## Competing interests

The authors declare no competing interests.
