## [Peer Review File · Nature Communications]

Reviewers' Comments:

Reviewer #1:

Remarks to the Author:

The authors have well addressed my (minor) comments, as well as the comments of the other referees. I agree with the authors that the points raised by referee 3 were minor, and find that the authors have written a high-quality reply to these points that has improved the paper. As such, I strongly recommend the paper to be published in Nature Communications. It is an original work that pushes our understanding of this intensely researched two-dimensional magnet. In my opinion, the work well meets the high quality and broad appeal required by Nature Communications.

Toeno van der Sar

Reviewer #2:

Remarks to the Author:

I sincerely appreciate the efforts made by the authors to address the comments by me and the other reviewers.

As the authors also mention in their reply, the manuscript provides a series of missing links in the understanding of the magnetic properties of atomically thin CrSBr that, although not being of groundbreaking novelty, will still be very interesting for a large community and will offer several perspectives for further studies.

For these reasons I would definitely consider the manuscript suitable for publication in Nature Communications in the current form.

If anything I would suggest the authors to review the revisions introduced against possible mistakes/repetitions (e.g. on page 1 it would be good to specify that the novelty is on the "nanoscale properties of THIN CrSBr", or on page 3 the authors might prefer to use "semiconductor magnets" instead of "semiconductor magnetics", or on page 5 avoid the repetition of "explore").

Reviewer #3:

Remarks to the Author:

I do not think the author's reply is convincing, particularly on two questions. First, even though some published papers in the community misused the concept of magnetic phase, this cannot justify it is scientifically correct to use it in that way. When an external magnetic field is applied to align all spin towards one direction, that is not "ferromagnetic phase" and does not have the ferromagnetic phase order of parameter. Going to an extreme, if a plastic bag is placed in a 10,000 Tesla magnetic field (assuming we can afford such a huge field), the spin in the plastic will be aligned along the same direction, but we cannot call this a "ferromagnetic phase". This is a textbook knowledge that the nature of the exchange interaction should be seriously considered when we discuss the magnetic phase.

The second point I am not satisfied with is that the authors present many different aspects which cannot coherently work together to deliver an in-depth understanding. I agree it can provide different aspects of understanding but cannot focus on depth. For example, if the air stability is the focus of this work, why air stability in this 2D magnet is possible and how this air stability can be engineered (just for example)? Addressing related questions can effectively deepen the depth. In

short, after reading the manuscript, the audience should feel they learned new knowledge and insights by the control experiments and progressive analysis into depth. Rather, the authors present different aspects, many of which are scattered and not related to air stability. Covering various aspects is not the true meaning of "broad audience", which I think the authors interpret wrongly.

Reviewer #1 (Comments for the Author):

The authors have well addressed my (minor) comments, as well as the comments of the other referees. I agree with the authors that the points raised by referee 3 were minor, and find that the authors have written a high-quality reply to these points that has improved the paper. As such, I strongly recommend the paper to be published in Nature Communications. It is an original work that pushes our understanding of this intensely researched two-dimensional magnet. In my opinion, the work well meets the high quality and broad appeal required by Nature Communications.

We thank the reviewer for this positive assessment and recommendation to publish our paper in Nature Communications

Reviewer #2 (Comments for the Author):

I sincerely appreciate the efforts made by the authors to address the comments by me and the other reviewers. As the authors also mention in their reply, the manuscript provides a series of missing links in the understanding of the magnetic properties of atomically thin CrSBr that, although not being of groundbreaking novelty, will still be very interesting for a large community and will offer several perspectives for further studies. For these reasons I would definitely consider the manuscript suitable for publication in Nature Communications in the current form. If anything I would suggest the authors to review the revisions introduced against possible mistakes/repetitions (e.g. on page 1 it would be good to specify that the novelty is on the “nanoscale properties of THIN CrSBr”, or on page 3 the authors might prefer to use “semiconductor magnets” instead of “semiconductor magnetics”, or on page 5 avoid the repetition of “explore”).

We thank the reviewer for this positive assessment and recommendation to publish our paper in Nature Communications.

We have implemented all of the remaining recommendations of the referee, including the explicit mention of “properties of thin CrSBr” (p.1), fixing the typo of “semiconductor magnetics” on p.3 and unnecessary repetitions of the word “explore” on p. 5.

Reviewer #3 (Comments for the Author):

I do not think the author’s reply is convincing, particularly on two questions. First, even though some published papers in the community misused the concept of magnetic phase, this cannot justify it is scientifically correct to use it in that way. When an external magnetic field is applied to align all spin towards one direction, that is not “ferromagnetic phase” and does not have the ferromagnetic phase order of parameter. Going to an extreme, if a plastic bag is placed in a 10,000 Tesla magnetic field (assuming we can afford such a huge field), the spin in the plastic will be aligned along the same direction, but we cannot call this a “ferromagnetic phase”. This is a textbook knowledge that the nature of the exchange interaction should be seriously considered when we discuss the magnetic phase. The second point I am not satisfied with is that the authors present many different aspects which cannot coherently work together to deliver an in-depth understanding. I agree it can

provide different aspects of understanding but cannot focus on depth. For example, if the air stability is the focus of this work, why air stability in this 2D magnet is possible and how this air stability can be engineered (just for example)? Addressing related questions can effectively deepen the depth. In short, after reading the manuscript, the audience should feel they learned new knowledge and insights by the control experiments and progressive analysis into depth. Rather, the authors present different aspects, many of which are scattered and not related to air stability. Covering various aspects is not the true meaning of “broad audience”, which I think the authors interpret wrongly.

We thank the reviewer for their renewed assessment of our work. We also thank them for the pictorial description of paramagnetism with their enlightening example of a plastic bag placed in a 10'000 T field. This picture is very helpful in explaining to the referee the difference between a paramagnetic phase and the spin-flipped phase we discuss in our work. As the referee may have noted, the field at which we observe the spin-flip, and the occurrence of a CrSBr bilayer with nearly fully aligned spins, is a factor of 50'000 smaller than the field given in the referee's example (0.2 T vs. 10'000 T). Applying textbook knowledge on the basic theory of paramagnetism to the case of the spin-3/2 system in CrSBr, at 0.2 T and $T \sim 4$ K, one finds a paramagnetic magnetisation of $\sim 1 \mu_B / \text{nm}^2$, which is a factor ~ 30 smaller than the magnetisation we observe in CrSBr bilayers in fields above the spin-flip transition. This order-of-magnitude discrepancy immediately shows that the spin-flipped state we observe cannot be called paramagnetic, as insinuated by the referee's plastic-bag example.

What we do agree with is that the interlayer exchange coupling remains antiferromagnetic in all cases (all while the intralayer exchange interaction is ferromagnetic). A small subset of readers might possibly be confused by the notion of a “ferromagnetic phase”, even if this terminology is widely used in literature and many textbooks and is certainly scientifically sound.

To address this minor source of confusion, we now amended our terminology in that we:

-More explicitly explain what we refer to in the context of the metamagnetic transition we observe and study, where on p.3 of the manuscript, we now write

«For magnetic fields of few 100 mT applied along the b-axis, CrSBr undergoes a metamagnetic transition from an AFM state, with low magnetization, to a state of strong magnetization...»

-For the remainder of the text, we then refer to “phases with zero and nearly saturated magnetization”.

While these modifications slightly affect the text's readability, they may avoid further misconceptions among some of our future readers.

We remain convinced that the broad range of aspects on nanomagnetism of few-layer CrSBr will be of interest and relevance to a broad readership and are glad to see that referees 1 and 2 share this opinion.